# Hydrochloric Acid Catalyzed Hydrothermal Treatment to Recover Phosphorus from Municipal Sludge

Kai Liu [1,†], Yang Xue [2,†], Yawei Zhai [2,*], Lisong Zhou [3] and Jian Kang [2]

[1]  School of Municipal and Environmental Engineering, Henan University of Urban Construction, Pingdingshan 467036, China
[2]  Faculty of Engineering, China University of Petroleum-Beijing at Karamay, Karamay 834000, China
[3]  Karamay Shuntong Environmental Technology Co., Ltd., Karamay 834000, China
*   Correspondence: zhaiyw@cupk.edu.cn
†   Co-first author: These authors contributed equally to this work.

**Abstract:** Resource utilization of sludge is critical because traditional sludge treatment methods cause a large amount of nutrient loss. This study investigated the impact of hydrochloric acid quantity, reaction temperature, and time on phosphorus release and migration from municipal sludge during hydrothermal treatment and designed a sludge disposal method for the recovery and utilization of phosphorus resources. We know that hydrochloric acid destroys the complexation of calcium and phosphorus precipitates, leading to the selective transfer of phosphorus to the liquid phase, and that the addition of 1–5% hydrochloric acid corresponds to a phosphorus extraction rate in the range of 0.3–98%. When hydrochloric acid is added, a change in temperature and reaction time has a negligible effect on phosphorus. Phosphorus can be recovered using the liquid product obtained under the optimal hydrothermal reaction conditions (adding 5% HCl at 205 °C for 30 min). After adjusting the pH value and adding the magnesium source, struvite ($MgNH_4PO_4 \cdot 6H_2O$) can be precipitated quickly and with high purity. At a cost of USD 27.8/ton of sludge, this method can recover 94% of the phosphorus in the sludge, and the bioavailable phosphorus ratio of the product is 93%, therefore, providing an important alternative to existing phosphorus recovery technologies.

**Keywords:** municipal sludge; hydrothermal carbonization; phosphorus; hydrochloric acid; struvite; recycling

## 1. Introduction

Humans are facing a serious "phosphorus challenge" because of the essentiality and non-renewable (phosphate ore) nature of P resources [1]. Phosphorus is the basic structural element of all biological cell membranes and plays an important role in life activities [2]. The known reserves of phosphate ore will likely be exhausted in 50–100 years based on supply and demand dynamics [3]. Such an imbalance between P demand and resources necessitates the development of different ways to reduce P mining.

In the European Union, more than 27% of imported P is lost in wastewater [4]. This phosphorus ends up in municipal sludge (MS), ranging from 1000 s mg/kg to several weight percent [5]. As the world population grows and urbanization increases, wastewater sludge poses several challenges, including overproduction and treatment requirements [6,7]. Landfills, incineration, agricultural applications, and other sludge treatment methods have limitations, such as the loss of nutrients, leakage of harmful elements, and excessive gas emissions [8–11]. Therefore, it is important to find alternative options to decrease sludge and recover P resources.

Recently, hydrothermal carbonization (HTC) has received increasing attention as a sustainable, economical, and efficient method for sludge treatment [12]. In the HTC process, biomass is heated in water at subcritical temperatures (180–240 °C) and autogenous

pressure to promote a series of complex reactions, such as hydrolysis, decarboxylation, polymerization, and aromatization of raw materials [13,14]. HTC treatment allows the organic matter in the feedstock to be dissolved in the liquid product (process water, PW), thereby effectively reducing the sludge volume; moreover, this process leads to PW, which contains a considerable amount of nutrients [15]. Simultaneously, a carbon-rich solid is generated, which is called hydrochar, and its main potential applications include solid fuel and soil remediation material [16,17]. By adjusting the hold time, temperature, and other process parameters, these reactions can be controlled to further influence the product properties.

Studies have highlighted that after HTC treatment, large amounts of P from the feedstock are distributed in the hydrochar owing to the large amount of metal co-precipitation with phosphate [18]. However, relevant amounts of phosphate are also found in PW under acidic conditions. For example, Ekpo et al. [19] extracted 94% of P from PW by adjusting the pH of the sludge slurry with $H_2SO_4$ before HTC treatment. In addition, the addition of inorganic acids allowed the release of more P at a lower acid concentration than when using organic acids as organic acids are more likely to break down and chemically react during the HTC process [20]. Although reaction temperature and time can also affect the behavior of P during hydrothermal carbonization [21], the addition of inorganic acids is the most significant factor determining the P distribution [19]. Direct extraction of P from hydrochar by acid leaching is another way of releasing it, which involves additional process conditions (extraction time, reagent concentration/volume) [3]. For instance, Lucian et al. devised an HTC-struvite phosphorus recovery process wherein acid was introduced to the product post-HTC treatment for extraction purposes, and the resulting extraction liquid was utilized for struvite crystallization, ultimately achieving a phosphorus recovery rate of 85% [22]. In addition, compared with methods such as anaerobic digestion and chemical leaching to release P, HTC is less restrictive in terms of environmental conditions and is more cost-effective [23,24].

After dissolving the phosphorus by hydrothermal carbonization, it is also necessary to choose a suitable method to recycle the phosphorus into the final product. Cell disintegration and matter decomposition accompanied by hydrothermal carbonization lead to high loads of ammonia in PW [25]. Combined with P leaching in liquid products and the characteristics of high ammonia content, the PW seems to be more suitable for P reclamation by struvite precipitation ($MgNH_4PO_4 \cdot 6H_2O$) [26]. Struvite presents an advantage in its availability as an agricultural fertilizer, providing P and N in an efficiently available form for plant nutrition [27]. Therefore, a cascade process combining acid-mediated HTC and struvite precipitation to recover P from the PW is an interesting approach; however, current research is mainly focused on the recovery of P from hydrochar. Simultaneously, due to limited data, a systematic study on the transformation of P by carbonization conditions has not been performed.

The purpose of this paper is to recover P from HTC PW. Specifically, appropriate P-rich PW is obtained by adjusting the reaction time, temperature, and acid concentration, and P is recovered by the precipitation of struvite. Preliminary assessment plans are devised to ascertain the economic costs to determine process feasibility. This study optimized the conditions of the entire P recovery process and provided a reference for practical production.

## 2. Results and Discussion

### 2.1. Extraction and Species of Phosphorus

2.1.1. Effect of Acid Addition

Figure 1 shows that the P content in HTC PW without acid is 7.45 mg/L, which is only 0.3% of the raw material P content. The extremely low P content is caused by metal ions, which are represented by $Ca^{2+}$. The high Ca content in MS can induce the complexation of phosphate anions (forming calcium-associated phosphate minerals). These phosphates exist in colloidal or free form and adhere to the surface or interior of the growing hydrothermal carbon, resulting in the formation of insoluble minerals with hydrochar precipitation during

the HTC process [28]. With the addition of 1–2% HCl, the extraction efficiency of P was not obvious. With further increases in the HCl concentration, P precipitates in hydrochar began to dissolve significantly. This critical phenomenon has been reported in related studies, and it mainly involves the conversion between apatite IP and non-apatite IP in hydrochar (before significant P dissolution) [29,30]. The addition of hydrochloric acid enhances the acidity of the system, which in turn impedes P complexation by reducing the solubility of the Ca-associated P precipitates [20], resulting in an increase in the amount of P in the PW (Figure 1b). This mechanism is confirmed by the high extraction rate of P with strongly acidic hydrothermal solution. The PW of the HTC treatment with 3% HCl added has a pH value of 1.42, while that with 5% HCl had a pH value of 0.53. At the same time, the extraction rate of P increased from 29% to 95%. The lowest pH also corresponds to the highest ammonia nitrogen concentration because stable peptide bonds in proteins can be catalyzed quickly in acidic or alkaline environments [31].

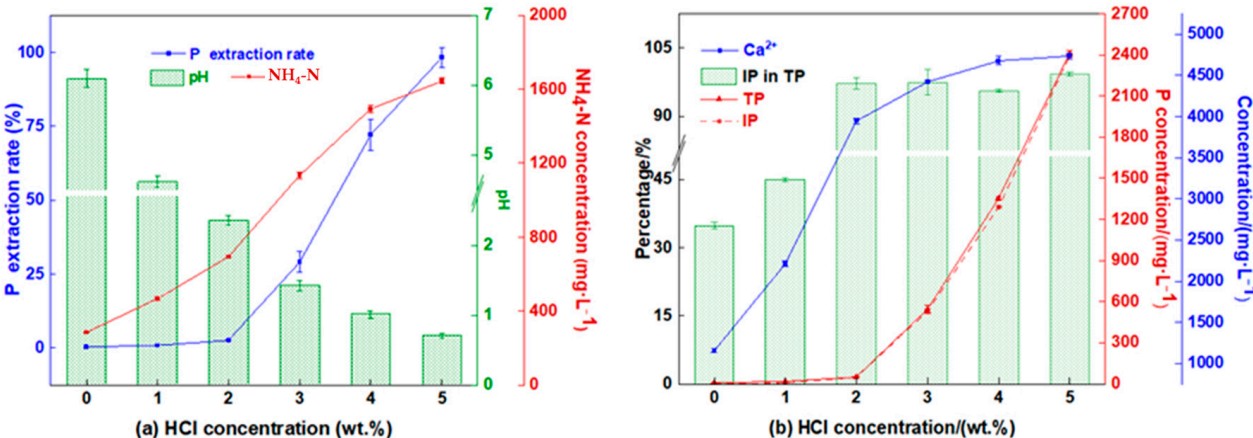

**Figure 1.** Extraction of phosphorus (**a**) and species of P in process water (**b**).

The positive effect of using HCl additives on the species of phosphate is obvious (Figure 1b). The IP proportion of the HTC PW with 2% HCl was 97.2%, which represented an increase of 1.79 times compared to the acid-free solution, and the maximum amount of IP proportion (99.2%) was observed in the aqueous phase with 5% HCl. The increase in TP concentration was mainly attributed to the increase in IP because the acid promoted the hydrolysis of OP during the HTC process [29]. At the same time, the IP content increased from 34.77 (0% HCl) to 44.9 (1% HCl) and 97.2 (2% HCl) as polyphosphate decreased at higher temperatures, which resulted in an increase in the amount of short-chain polyphosphate [32]. These results suggest that adding higher concentrations of HCl can promote the dissolution of P into PW and the conversion of organophosphorus to IP.

### 2.1.2. Effect of Temperature and Hold Time

Phosphorus in the sludge was selectively transferred to the PW under acidic conditions. Although struvite precipitation can be carried out directly under these experimental conditions, we continued to optimize the reaction temperature and time to reduce costs as much as possible while maintaining the P recovery.

Figure 2a shows the effect of temperature on P recovery. The P content of PW prepared at 205 °C was lower than that prepared at lower temperatures, which was likely a temperature-dependent effect. High temperatures may suppress the release of P into the liquid phase and generate abundant P in the solid phase [21]. From 75 °C to 205 °C, the extraction rate of P varies in the range of 15% because the dissolution efficiency of acid is higher than that of temperature, and the higher concentration of acid dissolves a certain amount of phosphate in the aqueous phase along with the decomposed organic compounds. When the temperature was lower than 175 °C, the significance of the temperature effects was stronger and the content of P in PW decreased obviously. Within the investigated

temperature range, IP was the main type of phosphate. The overall IP ratio fluctuated around 90%, and a decrease in temperature led to a slight decrease in the IP fraction (Figure 2a). This can be attributed to the hydrothermal carbonization process, which leads to the hydrolysis of organophosphorus to pyrophosphate and the gradual conversion to orthophosphate; a more thorough degree of reaction is realized at higher hydrothermal temperatures [30]. $NH_4$-N showed a clear decreasing trend as the temperature decreased (Figure 2b) because the continuous reduction in protein-N during HTC contributed to the formation of dissolved $NH_4$-N from amino acids through deamination and ring-opening reactions at higher temperatures [33]. A sufficient supply of energy breaks stable chemical bonds and forms dissolved ammonia nitrogen.

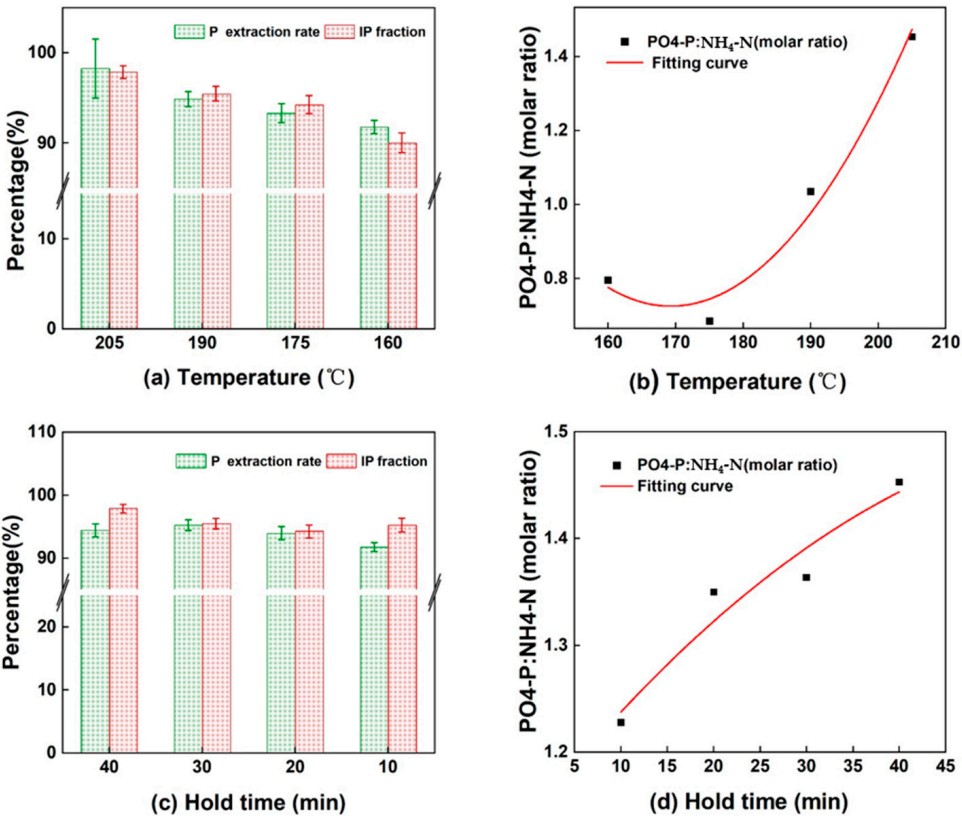

**Figure 2.** Effect of temperature on phosphorus extraction (**a**) and $PO_4$-P:$NH_4$-N molar ratio (**b**); effect of hold time on phosphorus extraction (**c**) and $PO_4$-P:$NH_4$-N molar ratio (**d**).

The effect of time on P recovery was studied by collating the collected data in the same manner as that when studying the effect of temperature (Figure 2c). Reaction time did not seem to have a significant effect on P allocation, at least under our study conditions. Similar phenomena were also reported by Zhao et al. [34], who showed that the optimal temperature range of HTC is 170 °C to 230 °C and the duration is 5 to 120 min, in which the processing time usually shows a limited effect. Figure 2d shows that the concentration of ammonia nitrogen increased slightly, and the interval of 10–40 min increased the concentration of ammonia nitrogen by approximately 20%. This is because N is mainly released in the form of organic N and then hydrothermally decomposes into $NH_4$-N, especially at higher temperatures and longer times [35]. The above results show a relatively rapid solubilization of P in the HTC PW.

The reaction temperatures of 170 °C to 205 °C and durations of 20 min to 40 min were further optimized to obtain the best reaction conditions (Figure 3). At higher temperatures (>195 °C), the extraction rate of P was higher than 95% and the proportion of IP and $NH_4$-N was maintained at a high level. To maintain a higher $PO_4$-P/$NH_4$-N ratio, the reaction time could be reduced by appropriately increasing the temperature. Therefore, the optimal

reaction conditions selected was 195 °C for 25 min. Under the optimized conditions, the concentrations of $NH_4^+$-N, $PO_4^{3-}$-P, and $Mg^{2+}$ in the PW were 1530 mg/L, 2379.5 mg/L (97.9% of total P), and 1030 mg/L, respectively.

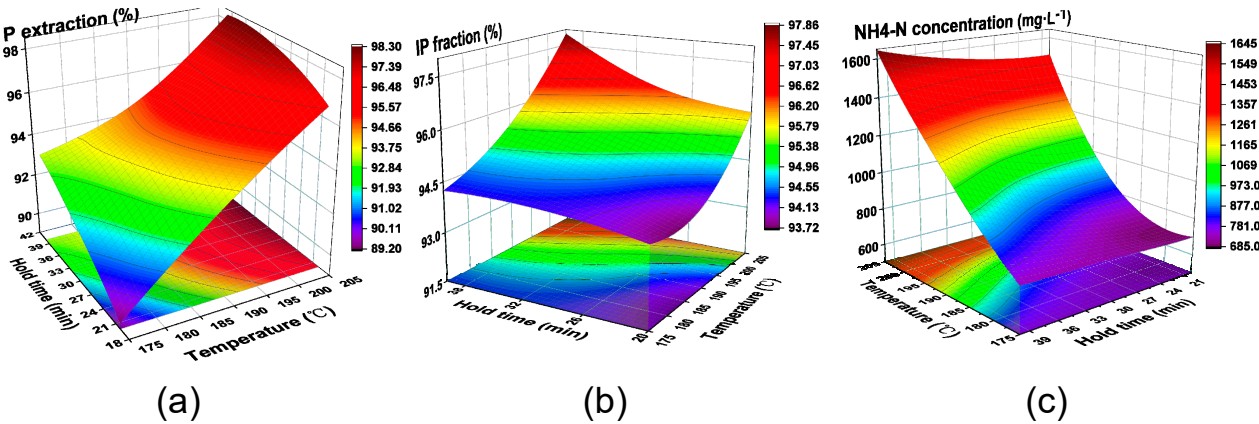

**Figure 3.** Optimization of P extraction (**a**), IP fraction (**b**), and $NH_4$-N (**c**).

## 2.2. Recovery of Phosphorus

Figure 4 shows the recovery efficiency of struvite precipitation under different pH conditions. It is generally believed that struvite is formed under alkaline conditions, and if the pH is too low, the phosphoric acid ions in the solution will protonate, eventually leading to the formation of $Mg(H_2PO_4)$. When the pH is too high, P is more inclined to form $Ca_5(PO_4)_3(OH)$. Therefore, the three pH values of 8, 9, and 10 were selected for the specific recovery process. During the experimental operation, the pH of all precipitates decreased slightly at the instant of particle formation, which is consistent with the dominant form of phosphate ($PO_4^{3-}$) present under these conditions. Under alkaline conditions, P recovery was always high (>85%), which can be attributed to the excess $Mg^{2+}$ and $NH_4^+$ promoting the formation of struvite. Struvite exhibits lower purity at high pH values (Figure 4) because other metal ions compete with $Mg^{2+}$, and as the pH increases, the association of precipitates with $Ca^{2+}$ increases, and hydroxide precipitates gradually form [36]. At pH = 8.9, the recovery rate of P reached 99%, indicating that struvite recovery of P is very effective. This relatively low pH environment avoided the volatilization loss of ammonia under strongly alkaline conditions.

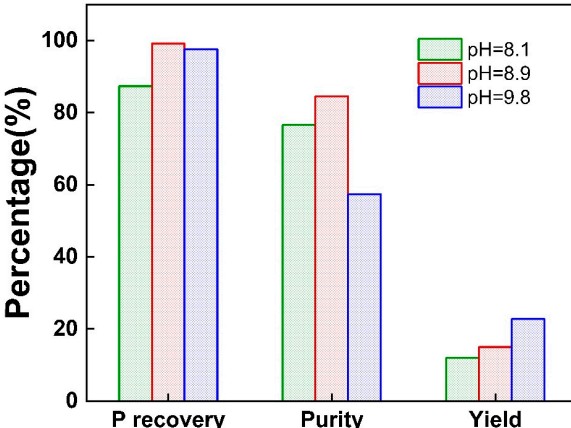

**Figure 4.** Recovery efficiency of struvite in different pH environments.

Similar results were reported in other studies. For example, Szogi et al. [37] extracted P directly from raw feces by chemical leaching and then recovered it into a concentrated solid form (calcium phosphate) by adding lime and organic polymer, and they finally achieved a

90% P recovery rate. Heilmann et al. [28] treated the solid product of HTC with acid, added alkali to the acid extract to reach a pH value of 9, filtered the main calcium phosphate, and achieved a final phosphate recovery of 80–90%. In addition, the biologically available phosphorus in struvite under optimal recovery conditions is 198.26 mg/g, accounting for 93% of total phosphorus. It indicates that the recovered struvite has potential as fertilizer. These results indicate that our method for P recovery is equally effective and has fewer steps.

### 2.3. Evaluation of Hydrochar

The dewatered sludge and optimized hydrochar were tested and analyzed (Table 1). Hydrochar had a low higher heating value (HHV), which indicated that it is not suitable for direct combustion for energy supply (general fuel HHV is about 15 MJ/kg). After the HTC treatment, the sludge material was transformed into black hydrochar, which showed a state of fine particles. The volatile content of hydrochar was very low, which was mainly caused by the decomposition of organic matter during the HTC process. In addition, the excess loss of volatiles led to an increase in fixed carbon and ash fractions. The decrease in the hydrochar yield was mainly due to the dissolution of the precipitation, which is demonstrated by the increase in the concentration of calcium ions in Section 2.1.1. At the same time, the effect of HTC treatment on the dissolution of inorganic salts in sludge was not obvious, and even when a higher concentration of hydrochloric acid was added, the total content of C, H, N, O, and S was only 34%. In addition, the H/C and (N + O)/C ratios of the product under the optimized conditions were lower than those without optimization. Therefore, compared with the raw material, the hydrochar product has a higher degree of aromatization and lower polarity [38], indicating that it has higher stability in soil and a lower adsorption capacity for polar substances [39].

**Table 1.** Properties of the tested samples.

| Samples | Proximately Analysis(wt.%) | | | Ultimately Analysis(wt.%) | | | | | HHV [b] (MJ kg$^{-1}$) | Solid Yield [c] | H/C | O/C | (N + O)/C |
|---|---|---|---|---|---|---|---|---|---|---|---|---|---|
| | Ash | Volatile | Fixed Carbon | C | H | N | S | O [a] | | | | | |
| Dewatered sludge | 55.48 | 41.17 | 3.34 | 13.38 | 3.08 | 1.12 | 3.61 | 23.33 | 5.06 | \ | 2.76 | 1.31 | 1.83 |
| Hydrochar (0%HCl-205 °C-40 min) | 57.31 | 36.12 | 6.57 | 11.79 | 2.06 | 0.44 | 3.72 | 24.69 | 3.15 | 53.75 | 2.09 | 1.57 | 2.13 |
| 1-205-40 | 63.47 | 27.87 | 8.66 | 12.58 | 2.41 | 0.42 | 4.22 | 16.9 | 4.56 | 45.99 | 2.30 | 1.01 | 1.83 |
| 2-205-40 | 63.03 | 24.63 | 12.34 | 14.01 | 2.58 | 0.41 | 4.46 | 15.51 | 5.44 | 36.74 | 2.21 | 0.83 | 1.51 |
| 3-205-40 | 64.72 | 20.8 | 14.48 | 15.37 | 2.45 | 0.39 | 3.73 | 13.34 | 5.88 | 29.88 | 1.91 | 0.65 | 1.19 |
| 4-205-40 | 65.31 | 20.11 | 14.58 | 16.55 | 2.31 | 0.38 | 2.99 | 12.46 | 6.13 | 27.63 | 1.67 | 0.56 | 1.03 |
| 5-205-40 | 20.01 | 66.13 | 13.86 | 17.01 | 2.05 | 0.37 | 1.90 | 12.54 | 5.85 | 23.54 | 1.45 | 0.55 | 1.01 |
| Optimized hydrochar (5-195-25) | 65.97 | 19.75 | 14.28 | 17.15 | 2.12 | 0.38 | 1.95 | 12.44 | 6.00 | 24.09 | 1.48 | 0.54 | 0.75 |

[a] By difference: 100 − C (wt.%) − H (wt.%) − N (wt.%) − S (wt.%) − Ash (wt.%). [b] HHV = 0.3491 C (wt.%) + 1.1783 H (wt.%) + 0.1005 S (wt.%) − 0.1034 O (wt.%) − 0.0151 N (wt.%) − 0.0211 Ash (wt.%) [25]. [c] Solid yield = 100*hydrochar quality/dewatered sludge quality.

### 2.4. Economic Outlook of the Process

Although struvite is economically viable and profitable for commercial applications, the successful use of the entire technology depends on the sustainability of the overall program economics. For hydrothermal processes, the acid cost is approximately USD 3.52/ton of wet sludge for 5% HCl and 1:10 biomass-to-water ratio (corresponding to 91% moisture content).

For the recovery process, the cost of NaOH (USD 365/ton, Cangzhou Runnuo Chemical Products Co., Ltd., Cangzhou, China) for pH adjustment was higher, approximately USD 9.6/ton, and the total cost of struvite recovery is calculated to be USD 10.2/ton of dehydrated sludge. With energy costs (USD 0.13/Wh, State Grid Corporation of China, Beijing, China), the entire process costs approximately USD 27.8/ton of wet sludge. When

MgO or other inexpensive alkaline chemicals are introduced into actual production, the cost is significantly reduced [40]. This process was more cost-effective than the method of [41], which released P by adding acidophilic heterotrophic iron-reducing (HIR) bacteria (EUR 114/ton dewatered anaerobic sludge), and other P removal processes, such as concentrated solutions or permeable membranes [42].

## 3. Materials and Methods

### 3.1. Municipal Sludge

The municipal sludge used in this study was withdrawn from a local sludge disposal plant in Xinjiang, China. For characterization and storage, the sludge was dried at 105 °C to a constant weight and then ground to a final particle size smaller than 0.15 mm. The sludge was stored in plastic bags (4 °C) prior to the HTC experiments. The proximate analyses were performed by measuring the mass loss at varying temperatures in accordance with GB/T 212-2008 [43], while the ultimate analyses were conducted using an elemental analyzer (DHF82; Elementar, Berlin, Germany). The raw sludge moisture content was 92.35%. The phosphorus content of dewatered sludge was 25.51 mg/g, and other characteristics are shown in Table 1.

### 3.2. Hydrothermal Carbonization of Sludge

The hydrothermal carbonization reaction was performed in a batch reaction kettle (75 mL) coated internally with polytetrafluoroethylene and equipped with an automatic temperature controller and pressure gauge. The reactor was charged with 4 g of dried sludge with either 40 mL of deionized water or acid solution. The time was started after heating to the specified temperature. As the temperature rose, the pressure inside the reactor gradually increased, and the water in the reactor would be in a subcritical state. When the duration was reached, the power was turned off at the stove and the solution was allowed to cool naturally to room temperature (25 °C). The hydrochar and PW were separated by vacuum filtration. The collected hydrochar was subjected to drying at 105 °C until reaching a stable weight. The samples were stored in the refrigerator at 4 °C until testing. Three parameters were considered when assessing the HTC process: HCl concentration, reaction temperature, and hold time. These parameters were optimized individually. The initial temperature was 205 °C, and the initial hold time was 40 min.

### 3.3. Characterization of Product

The contents of inorganic phosphorus (IP) in the product supernatant were detected by the molybdenum blue method, and the samples were digested with $K_2S_2O_4$ (5.0%, wt.%) for 30 min at 121 °C in an autoclave pot. Total phosphorus (TP) was determined with the IP method. The organic phosphorus (OP) content is the difference between the IP and TP content. The absorbance of the solution was measured at a wavelength of 700 nm using a UV-visible spectrophotometer (N4, Shanghai Yuan Analysis Instrument Co., Ltd., Shanghai, China). The ammonium concentration was determined by measuring the absorbance after adding Nessler reagent (BKMAN, Shanghai, China). pH was measured using a digital pH meter (PHS-3C, Shanghai Yuan Analysis Instrument Co., Ltd., Shanghai, China). Calcium and magnesium ion concentrations were determined using ion chromatography. Each experiment was conducted in triplicate, and the average of the results was used as the final determination. For the determination of bioavailable phosphorus [30], 0.2 g of sample was first added into 20 mL of NaOH (1 mol/L) and then added into 8 mL HCl (3.5 mol/L) after shaking for 16 h. The phosphorus content was determined with the IP method.

### 3.4. P Recovery via Struvite Precipitation

Struvite was crystallized in a beaker (250 mL) with the HTC PW. The initial $PO_4^{3-}:NH_4^+$ ratio of the optimized PW was 1:1.5. A certain amount of $MgCl_2 \cdot 6H_2O$ was added to the solution to ensure that the molar ratio of $PO_4^{3-}:Mg^{2+}$ was 1.5. The precipitation reaction was performed according to Equation (1). NaOH (4 mol/L) was added to adjust the alka-

linity, and the mixture was increased to 100 mL before the pH adjustment. Stirring was performed using a magnetic agitator at a rotating speed of 100, and the reaction time was 30 min. The evaluated outputs include the efficiency of P recovery ($P_{Rec}$) and purity of the struvite fraction in the product (struvite purity, SP) [30]. Struvite yield is expressed as a percentage (15% represents 0.15 g struvite/g sludge).

$$Mg^{2+} + NH_4^+ + PO_4^{3-} + 6H_2O = MgNH_4PO_4 \cdot 6H_2O \tag{1}$$

$$P_{Rec}(\%) = \frac{C_{PO_4^{3-}}^{before} - C_{PO_4^{3-}}^{after}}{C_{PO_4^{3-}}^{before}} \times 100 \tag{2}$$

$$SP(\%) = \frac{n_N \times M_{struvite}}{m_{struvite}} \times 100, \tag{3}$$

where $C_{PO_4^{3-}}^{before}$ is the concentration of $PO_4{}^{3-}$ before the reaction, $C_{PO_4^{3-}}^{after}$ is the concentration of $PO_4{}^{3-}$ after the reaction, $n_N$ is the mole number of nitrogen, $M_{struvite}$ is the molar mass of struvite, and $m_{struvite}$ is the mass of the precipitate.

*3.5. Economic Analysis*

A preliminary economic study was conducted on the overall process of phosphorus recovery, aiming to evaluate its economic feasibility. The study took into consideration the costs of energy and reagents, which are calculated as the product of consumption and price. The reagent cost includes three components: HCl (used for phosphorus release), NaOH (used for pH adjustment), and $MgCl_2$ (used for struvite coagulation). The consumption of reagents was directly measured through experiments.

The energy consumption (kJ/kg), obtained by converting the heat consumption into electrical energy, is expressed in Equation (4). With regard to the heat cycle of an actual project, the energy requirements of the process can be optimized by preheating the inlet temperature to 383 K. In the equation, $m_{DS}$ and mL represent the mass of dry sludge and liquid (water), respectively. $C_{P,DS}$, and $C_{P,L}$ stand for the specific heat of dry sludge (2.61 kJ/kg·K, experimentally measured) and water (4.18 kJ/kg·K), respectively.

$$E_{consumption}(KJ/KG) = [m_{DS} \cdot C_{P,DS} + m_L \cdot C_{P,L}] \cdot (T_{HTC} - 383) \tag{4}$$

**4. Conclusions**

In this study, acid HTC was used to treat municipal sludge, and 95% P was extracted into PW by adjusting the HCl concentration, temperature, and hold time. Almost all dissolved P was recovered in precipitates, and the purity of struvite exceeded 80%. The low cost of this scheme indicates a direction for the sustainable use of sludge. In addition, the drying in this study was only to facilitate the storage and treatment of the sludge (such as crushing and adjusting the ratio of biomass to water), so experimental results were not fully representative of industrial results (hydrothermal carbonization technology does not require dry pretreatment) [44]. It is also a direction to use hydrochar as a heat source for the reaction after modification (such as changing the calorific value and metal content to obtain better fuel characteristics).

**Author Contributions:** K.L.: writing; Y.X.: writing; Y.Z.: validation; L.Z.: formal analysis; J.K.: resources. All authors have read and agreed to the published version of the manuscript.

**Funding:** This work was supported by the Natural Science Foundation of Xinjiang Uygur Autonomous Region (2021D01F37), The Karamay Innovative Environment Construction Plan Project (2023hjcxrc0100), and the Project of Cooperation between Polytechnic and Enterprises in Karamay (XQZX20220070).

**Data Availability Statement:** Data are contained within the article.

**Conflicts of Interest:** Author Lisong Zhou was employed by the company Karamay Shuntong Environmental Technology Co., Ltd. The remaining authors declare that the research was conducted in the absence of any commercial or financial relationships that could be construed as a potential conflict of interest.

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
