# Peer review of "Hydrochloric Acid Catalyzed Hydrothermal Treatment to Recover Phosphorus from Municipal Sludge"

_catalysts, doi:10.3390/catal14010065_

Round 1

Reviewer 1 Report

Comments and Suggestions for Authors

1.       Line 91: “…the sludge was dried at 25°C for 72 h...”

Was the drying carried out in the air or by means of vacuum drying? What was the water/solids content of the raw sludge and was it possible to completely remove the water from the dried sludge using this process or did a residual moisture content remain?

2.       Line 100: “The reactor was charged with 4 g dried sludge with either 40 mL of deionized water or acid solution. “

The reactor was therefore filled to approx. 16% with the suspension. Was the atmosphere in the reactor filled with an inert gas or was it air? If it was air, does the oxygen in it have a relevant influence on the reaction/results?

3.       Line 105: “The slurry, including the hydrochar and PW, was separated by vacuum filtration. “

After the separation of process water and hydrochar, was the hydrochar dried in the same way as the sludge for further analysis?

4.       Line 255: “The raw sludge and optimized hydrochar were tested and analyzed (Table 1). “

I assume that all samples (sludge and hydrochar) were analyzed in a fully dewatered state?

5.       Line 259: “In addition, the excess loss of ash leads to an increase in fixed carbon and ash contents.”

Loss of ash leads to an increase in ash content? Please elaborate or correct.

6.       Line 268: table

Possibly add the solid yield of the hydrochar if data is available.

Comments on the Quality of English Language

 Minor editing of English language required

Reviewer 2 Report

Comments and Suggestions for Authors

Herein, I submit my comments for initial submission of the manuscript entitled: “Hydrochloric acid catalyzed hydrothermal treatment to recover phosphorus from sludge”.

Reuse of residual sludge is important because of the phosphorous releases into residual wastewater, especially considering that in the European Union, more than 27% of imported P is lost in wastewater. In order to present an alternative to recover the phosphorous from wastewater the authors evaluated the effect of hydrochloric acid quantity, reaction temperature, and time on phosphorus release and migration from a local sludge disposal plant in Xinjiang (China) during hydrothermal treatment and designed a sludge disposal method for the recovery and utilization of phosphorus resources.

I consider this manuscript suitable for publication. However, some ambiguities arose during the reading. Therefore, I came to a conclusion to encourage you to review it again. Please, use my comments listed below:

Comment 1: I would like to know if the sludge was characterized prior the experiments? If so, please, report the characterization, in order to know the amount of P in it.

Comment 2: Please, discuss the scalability of the process, because the authors mention the cost of HCl in ton, nevertheless the experiments were lab scale. Also, it would be interesting to see a block diagram with flows and the time residences, considering the process is batch.

Comment 3: Enhance the conclusion, stating future path.

Reviewer 3 Report

Comments and Suggestions for Authors

General comments:

In this manuscript authors report optimization of P recovery from mucinipal sludge via HCl acidification of the reaction mixture during HTC. They investigate the influence of HCl initial concentration, reaction temperature and residence time and they conclude that HTC process temperature around 200 °C and reaction holding time of 40 min (with HCl of 5%) are the optimal conditions to recover up to 95% by mass of the initial P with a 80% of purity… The work is in line with Catalyst jounal aims, the reasearch topic of high interest for the scientific community, due to the critical importance of P element especially in Europe. Despite the high relevance of the topic the experimental approach of the study must be better clarified in the manuscript text and relevant literature included in partcular for the possible scale up of the process at industrial level.

For waht stated above the present manuscript could be reconsidered for publication after the following specific comments will be properly addressed

Specific comments:

Title: the term “sludge” is too generic, it would be better to specify what kind of sludge in object of the present study, and thus changing the title appropriately.

Introduction:

in a relevant recent literature work, phosphorous reclamation, from agro-industrial sludge, has been investigated by Lucian and co-workers (Lucian, M. et al., Industrial-Scale Hydrothermal Carbonization of Agro-Industrial Digested Sludge: Filterability Enhancement and Phosphorus Recovery, Sustainability, 2021, 13, 9343. https://doi.org/10.3390/su13169343), in a continuous operating industrial scale HTC plant. They found that struvite and heavy metals can be recovered via acidic treatment of the recoverd hydrochars and HTC slurry obtained also at mild conditions (190 °C and 1 h of reaction residence time). Please discuss the relevance of this study for your work.  

Materials and Methods

The authors should explain and justify the need of drying the sludge before HTC treatments. It is has been repoted in the literature (Volpe, M. et al., Potential pitfalls on the scalability of laboratory-based research for hydrothermal carbonization, Fuel, 315, 2022, 123189, doi: 10.1016/j.fuel.2022.123189), that drying pretreatments of organic wet waste and sludge could alter the nature of the feedstock and thus HTC procces results could be not very well representative of the real process at the industrial scale.

Section 2.2 should be rewritten, the text is not very clearly expressed: what do you exactly mean with the following expression? “During the holding time, the pressure in the reactor was maintained above the water vapor pressure.” The pressure in the batch reactor would be the sum between the water vapor pressure and the gaseous phase produced during the reaction… please explain….

The following expression need to be rewritten: “The slurry, including the hydrochar and PW, was separated by vacuum filtration.” As for example, ….solid hydrochar was recovered via vacuum filtration of the HTC slurry…. “ you do not separate the slurry but the solid residue from the PW….

The definition of the SP facto (Eq. 3)r is not very clear to the reviewer… author should better explain how they can sbustract a quantity expressed in moles nM with a quantity expressend in grams/mole Mstruvite.

Results and discussion

Line 230: magnesium ions or ammonium ions?

Lines 258-9: what id the meaning of the following sentence? : “. In addition, the excess loss of ash leads to an increase in fixed carbon and ash contents.”

Line 278: 0.13 $/kWh (watt uniti s “W” not “w”) please leaves always a space between the numeric value and its units… (check all the manuscript text and correct accordingly).

For the economic outlook of the process … authors should indicate appropriate fonts for the price of chemicals and electric energy…. What happen if we use hydrochar as a possible source of thermal energy in the process? Ids the so obtained hydrochar suitable for combustion? What is its lower calorifc vale? (you can estimate by its elemental composition)

Comments on the Quality of English Language

Minor English language revision is needed.

Round 2

Reviewer 3 Report

Comments and Suggestions for Authors

The authors have correctly addressed all the reviewer's comments. The manuscript in now suitable for publication as it is.